# Examination of the Virome of Taro Plants Affected by a Lethal Disease, the Alomae-Bobone Virus Complex, in Papua New Guinea

**DOI:** 10.3390/v14071410

**Published:** 2022-06-28

**Authors:** Alejandro Olmedo-Velarde, Jarin Loristo, Alexandra Kong, Philip Waisen, Koon-Hui Wang, John Hu, Michael Melzer

**Affiliations:** 1Department of Plant and Environmental Protection Sciences, University of Hawaii, Honolulu, HI 96822, USA; jloristo@hawaii.edu (J.L.); atk412@hawaii.edu (A.K.); koonhui@hawaii.edu (K.-H.W.); johnhu@hawaii.edu (J.H.); 2Agriculture Department, The Papua New Guinea University of Technology, Lae 441, Papua New Guinea; pwaisen@ucanr.edu; 3Agriculture and Natural Resources Division, University of California, Indio, CA 92201, USA

**Keywords:** *Colocasia esculenta*, alomae-bobone virus complex, cytorhabdovirus, badnavirus

## Abstract

Alomae-bobone virus complex (ABVC) is a lethal but still understudied disease that is limited to the Solomon Islands and Papua New Guinea. The only virus clearly associated to ABVC is Colocasia bobone disease-associated virus (CBDaV). Taro (*Colocasia esculenta*) plants with and without symptoms of ABVC disease were sampled from two locations in Papua New Guinea and examined for viruses using high-throughput sequencing (HTS). Similar to previous reports, isolates of CBDaV were present only in symptomatic plants, further supporting its role in the disease. The only other viruses consistently present in symptomatic plants were badnaviruses: taro bacilliform virus (TaBV) and/or taro bacilliform CH virus (TaBCHV). If ABVC requires co-infection by multiple viruses, CBDaV and badnavirus infection appears to be the most likely combination. The complete genomes of two isolates of CBDaV and TaBCHV, and single isolates of TaBV and dasheen mosaic virus, were obtained in this study, furthering our knowledge of the genetic diversity of these relatively understudied taro viruses. HTS data also provided evidence for an agent similar to umbra-like viruses that we are tentatively designating it as Colocasia umbra-like virus (CULV).

## 1. Introduction

Taro (*Colocasia esculenta* [L.] Schott) is a major staple crop grown in tropical regions around the world. In many Pacific Island countries, taro has both economic and cultural importance. Taro is primarily propagated by vegetative cuttings, making it prone to the accumulation of viruses and the negative impacts resulting from their infection. A severe disease of taro associated with viral infection has only been reported in the Solomon Islands [1] and Papua New Guinea (PNG) [2]. Collectively known as the alomae-bobone virus complex (ABVC), disease symptoms include severe plant stunting, leaf mosaic and deformation, and chlorotic streaking of the petioles. Galls on the petioles are also reported (Figure 1A). Plant death or recovery depends on the plant age or variety of taro affected [1,2,3,4]. Despite the severity of ABVC, the limited geographic distribution of this disease has minimized its impact on global taro production.

ABVC has been associated with co-infection by multiple viruses [1,2,4]. After electron microscopy, ABVC was initially associated with infection by two types of bacilliform particles, a large rhabdovirus particle (provisionally designated Colocasia bobone disease virus, CBDV) and a small bacilliform particle with similarity to that of a badnavirus [4]. Subsequent studies have also identified similar particles in diseased plants [1,2,5] as well as flexuous rod-shaped particles [1]. Later, it was demonstrated that ABVC can be spread by the taro planthopper (*Tarophagus* spp.) [6]. Viruses that correspond to the observed particles in ABVC-affected plants have since been described: two rhabdoviruses, taro vein chlorosis virus (TaVCV) [7], a member of the species *Alphanucleorhabdovirus colocasiae*, and Colocasia bobone disease-associated virus (CBDaV) [8], a member of the species *Cytorhabdovirus colocasiae*; two bacilliform-shaped badnaviruses, taro bacilliform virus (TaBV) [9] and taro bacilliform CH virus (TaBCHV) [10], members of the species *Taro bacilliform virus* and *Taro bacilliform CH virus*, respectively; and one flexuous rod-shaped potyvirus, dasheen mosaic virus (DsMV) [11], member of the species *Dasheen mosaic virus*. Revill and collaborators [12] used PCR-based virus-specific assays to determine the virus status of 21 plant samples collected from PNG and the Solomon Islands with ABVC. Although CBDaV (then called CBDV) was found associated with ABVC plant samples from both Pacific Island countries, no PCR primer sequence information was provided. Still, such assays have the potential for type II errors (false negatives) if the primers used in these assays cannot detect all sequence variants. This is of particular concern for viruses that no sequence information is available or have a limited number or geographic distribution of isolates [13,14].

In understudied viral pathosystems where new species or genetic variants are likely, high-throughput sequencing (HTS) has emerged as a powerful tool for virome investigation [14,15], including rhabdoviruses [16]. In their characterization of CBDaV, Higgins and collaborators [8] performed HTS on a taro plant with symptoms of bobone whose progenitor originated from the Solomon Islands, but no other viral agents, if present in the plant, were reported. To provide a genetic context to the complex of virus particles previously observed in plants afflicted with ABVC, we used HTS to examine the virome of taro plants from PNG with and without symptoms of ABVC. From these sequence data and confirmatory testing using molecular diagnostics, we were able to identify the virus combinations present in ABVC-symptomatic plants, and partially characterize a new umbra-like virus infecting taro.

## 2. Materials and Methods

### 2.1. Plant Material

In June 2019, three leaf samples were collected from taro plants growing in Popondetta (two plants; 8°46′09.8″ S 148°14′16.8″ E) and Kokoda (one plant; 8°54′36.6″ S 148°00′08.4″ E), PNG, that were displaying typical ABVC symptoms (Figure 1A). Two additional leaf samples were collected from two taro plants growing in Girua (8°50′21.3″ S 148°15′37.9″ E) with no ABVC-related symptoms (Figure 1B). Tissue samples were placed in 15 mL conical tubes, submerged in RNAlater solution^®^ (ThermoFisher, Waltham, MA, USA), and transported to the University of Hawaii under USDA permit P526P-17-03915. Upon arrival, the samples were stored at −20 °C until processed.

### 2.2. HTS Analyses and Genome Sequencing

Double-stranded RNAs (dsRNAs) are an optimal and cost-effective starting material for HTS with the highest potential for detection and characterization of plant viruses and viroids [17]. Therefore, dsRNAs were extracted from approximately 5 g of pooled ABVC-symptomatic leaf tissue using either CF-11 (Whatman, Maidstone, UK) or C6288 (Sigma, St. Louis, MO, USA) cellulose chromatography [18]. These dsRNAs were used as a template to obtain randomly amplified cDNAs [19] and prepared for HTS using a Nextera XT DNA Library Prep kit (Illumina, San Diego, CA, USA). HTS was performed on an Illumina MiSeq platform (2 × 300 bp) at the University of Hawaii Advanced Studies in Genomics, Proteomics, and Bioinformatics (ASGPB) laboratory. Genome assembly and bioinformatic analyses were performed as described [20]. Briefly, paired-end reads were trimmed, and quality filtered using Trimmomatic 0.35 [21]. Then, the sequence reads were mapped to the reference genome of *C. esculenta* (PRJNA587719) using Geneious mapper plug-in implemented in Geneious v. 10.1.3 [22] to enrich sequence reads related to viruses. Trinity 2.2.0 [23] produced *de novo*-assembled contiguous sequences (contigs) from the non-taro sequence reads. These contigs were annotated using BLASTX searches [24] against the viral genome database (ftp://ftp.ncbi.nih.gov/genomes/Viruses/all.fna.tar.gz) last accessed on 20 March 2020. Contigs with similarities to virus sequences were then used as references for an iterative mapping approach [25] using the Geneious mapper and raw reads.

For the viruses detected, a group of overlapping primer sets (Appendix A) were designed based on the virus contigs and used to validate the HTS output and bridge sequence gaps by RT-PCR using either dsRNAs or total nucleic acids (TNAs) extracted from individual samples as detailed below. Poly-A tailed dsRNAs were generated using *E. coli* Poly (A) Polymerase (New England Biolabs, Ipswich, MA, USA) to determine both 5′ and 3′ ends. All amplicons were cloned into pGEM-T Easy (Promega, Madison, WI, USA) and three to five clones were sequenced per amplicon.

### 2.3. Virus Detection Using PCR-Based Assays

TNAs were extracted from ~0.1 g of leaf tissue from each sample using the CTAB method [26]. TNAs underwent cDNA synthesis using random primers, oligo-dT primer, and M-MLV reverse transcriptase (Promega, Madison, WI, USA) or Maxima-H minus reverse transcriptase (ThermoFisher, Waltham, MA, USA) using the manufacturers’ protocol. Conserved genomic regions along multiple nucleotide sequence alignments were identified using virus isolates available in GenBank and the isolates generated in this study. Virus-specific primers were designed as detailed previously [27]. RT-PCR and PCR assays were performed using virus-specific primer sets for CBDaV, TaBV, TaBCHV, taro reovirus (TaRV), DsMV, and TaVCV (Appendix A). Additionally, DNase-RT-PCR was used to further confirm the presence of episomal TaBV and/or TaBCHV. Briefly, TNA was DNase-digested with RQ1 RNAse-free DNase (Promega, Madison, WI, USA). The resulting DNase-digested TNA was then used in cDNA synthesis and RT-PCR assays as detailed above. Amplicons were gel extracted, purified, and bi-directionally sequenced or cloned into pGEM-T Easy with three clones sequenced per amplicon.

### 2.4. Rolling Circle Amplification

TNAs (not digested with DNase) were subjected to rolling circle amplification (RCA) using TempliPhi 100 Amplification Kit (Sigma-Aldrich, St. Louis, MO, USA). Amplification was performed at 36 °C for 18 h, then the reaction was terminated by incubation at 65 °C for 10 min. RCA-amplified DNA was independently digested using *Eco*RI and *Sac*I (ThermoFisher, Waltham, MA, USA) and analyzed by gel electrophoresis in 0.8% agarose gels in TBE buffer. Both restriction endonucleases were predicted to cut at least once in the TaBV and TaBCHV isolate sequences available at GenBank that were analyzed.

### 2.5. In Silico Genomic and Proteomic Analyses

Genomic and proteomic analyses were performed in silico as detailed previously [27]. Briefly, the NCBI ORFfinder program [28] was used to identify putative open reading frames (ORFs). BLAST searches and LALIGN [29] were used for pairwise sequence comparisons using nucleotide and inferred protein sequences and their respective orthologous sequences retrieved from GenBank.

### 2.6. Phylogenetic Analyses

Phylogenetic relationships between the taro viruses characterized in this study and their homologs were inferred using the nucleotide or amino acid sequences of whole genomes or multiple protein domains conserved among some or all of these viruses using the best model of evolution as determined in MEGA 7.0.26 [30]. Multiple sequence alignments were performed in MEGA 7.0.26 [30] using ClustalW algorithm [31]. Ambiguous positions for all the alignments were curated using Gblocks 0.91 b [32].

## 3. Results

### 3.1. Symptomology

The three taro plants from Popondetta and Kokoda with symptoms of ABVC were severely stunted (Figure 1A). Their small, misshapen leaf laminas were rugose with enlarged veins and were often chlorotic or necrotic. Leaf petioles were also enlarged and had a sunken, necrotic vasculature. Galling and enations were also present on some petioles. In contrast, the two taro samples collected from Girua had none of these major symptoms, but their leaves did show symptoms of interveinal chlorosis (Figure 1B).

### 3.2. Virome Analysis

HTS generated ~14.5 M paired-end reads (300 nt) from a cDNA library using dsRNA as a template which was extracted from pooled tissue sampled from the three ABVC-symptomatic plants (Figure 1A). Of these, ~8.3 M reads that did not map to the *C. esculenta* reference genome underwent *de novo* assembly, producing 663 contigs. Of these, 188 contigs were determined to be of viral origin (Table 1). In addition to contigs showing similarity to CBDaV (61 contigs), TaRV (55 contigs), TaBV (16 contigs), DsMV (18 contigs) and TaBCHV (37 contigs), a contig with similarity to umbra-like viruses such as sugarcane umbra-like virus (SULV), and opuntia umbra-like virus (OULV) was also found. The presence of this contig indicated a novel agent that we tentatively designate Colocasia umbra-like virus RNA (CULV). No sequences with similarity to TaVCV were identified in the HTS data.

PCR-based assays were used to determine the virus status of the samples collected from the three taro plants affected by ABVC as well as the two plants with interveinal chlorosis but no ABVC-associated symptoms (Table 2, Appendix A). Sequencing of the PCR amplicons was used to confirm >99% nucleotide identity to the sequence of the original virus contigs obtained using bioinformatics. Infection by CBDaV only was unique to samples collected from all three ABVC-affected plants. Samples from the two ABVC-affected plants from Popondetta were co-infected with TaRV, with all 10 genomic segments detected in both samples while the sample collected from the ABVC-affected plant from Kokoda was co-infected with DsMV and CULV. Both PCR and DNase-RT-PCR assays indicated the presence of the two badnaviruses, TaBV and TaBCHV, in samples from plants both symptomatic and asymptomatic for ABVC, with TaBCHV found in all plants sampled for this study (Table 2). RCA detected circular DNA molecules in all five samples, and bioinformatic analyses of the HTS data also indicated intact, circular TaBV genome (7825 nt). The intact TaBV genome sequence was generated as a circular scaffold after an iterative mapping approach (Appendix A). Together, these results indicated the presence of episomal forms of these viruses. As with the HTS data, TaVCV was not detected in any of the plant samples.

### 3.3. Molecular Characterization of Viruses Infecting Taro Plants from PNG

The complete genomes of two isolates of CBDaV infecting ABVC-affected taro plants were obtained using HTS data, followed by RT-PCR to bridge sequence gaps and determine terminal sequences. One isolate originating from Popondetta (CBDaV-PNG-P; Genbank ON086740) and one from Kokoda (CBDaV-PNG-K; ON086741). CBDaV-PNG-P and CBDaV-PNG-K were 12,205 nt in length (184K reads, 2296× coverage) (Table 1), shared 98.9% nt identity, and had the same genome organization. CBDaV-SI, the only other isolate of the virus sequenced to date and originating from the Solomon Islands [8], shared 82.9 and 94.7% identity in nucleotide and protein sequences respectively with the PNG isolates of the virus (Figure 2A). A notable feature not reported in the description of CBDaV-SI was a small open reading frame (ORF) putatively encoding a protein of unknown function between the glycoprotein (G) and polymerase (L) ORFs of all three isolates (Figure 2A). In a phylogenetic tree inferred from the whole genome sequences of CBDaV and their homologs and using the best model of DNA evolution (LG + G + I), CBDaV-PNG-P and CBDaV-PNG-K formed a monophyletic clade with CBDaV-SI and appeared most closely related to papaya virus E isolates of *Cytorhabdovirus caricae*. Similar to other cytorhabdoviruses, the CBDaV isolates had enough genetic divergence that different strains could be distinguished (PNG and SI) (Figure 3A).

Two badnaviruses, TaBV and TaBCHV, were identified in taro samples collected from PNG. The complete genomes of one isolate of TaBV and two isolates of TaBCHV were obtained using HTS data, RT-PCR, and RCA. TaBV-PNG-K (GenBank ON086737) was sequenced from a sample from Kokoda and found to be 7825 nt in length (2.5 M reads, 137,637× coverage) (Table 1). It had a genome organization identical to other isolates of the virus (Figure 2B). Of the fully sequenced isolates of TaBV in GenBank, it was most closely related to TaBV-Ug 75, with nucleotide and protein identities both under 90% (Figure 2B). However, partially sequenced isolates of the virus, including those from the Pacific Island countries of New Caledonia (NC1; AY186614, AY187267), the Solomon Islands (SI2; AY187266), and PNG (e.g., CE/PNG/105; MW651832) had much higher (>98%) nucleotide sequence identities and closer phylogenetic relationships with TaBV-PNG-K (Figure 3B). Isolates TaBCHV-PNG-K (GenBank ON086738) and TaBCHV-PNG-P (GenBank ON086739) were sequenced from the Kokoda and Popondetta samples, respectively. The genomes of TaBCHV-PNG-K and TaBCHV-PNG-K were 7769 and 7770 nt in length (2 K reads, 24× coverage) (Table 1), respectively, and shared 96.3% nucleotide identity and an identical genome organization. Both isolates were most similar to TaBCHV-Ke43 from Kenya (GenBank MG017325) with TaBCHV-PNG-K having nucleotide and protein identities respectively of 97.6 and 94.9% with the Kenyan isolate (Figure 2C). Phylogenetic analyses placed TaBCHV-PNG-K among African isolates of the virus (Figure 3C and data not shown). Both phylogenies of TaBV and TaBCHV were inferred using the best model of DNA evolution (GTR + G + I).

The complete genome of a DsMV isolate was recovered from a taro sample from Kokoda. Excluding the 3′ poly(A) tail, DsMV-PNG-K (GenBank ON086743) was 10,066 nt in length (3.9 K reads, 46× coverage) (Table 1) and possessed a single large ORF encoding a polyprotein typical of DsMV and other potyviruses (Figure 2D). DsMV-PNG-K was most similar to DsMV-I (GenBank KY242358) that was sequenced from taro in Hawaii. The two isolates shared nucleotide and protein identities of 82.3 and 89.9%, respectively. In a phylogenetic tree inferred from DsMV, whole genome sequences and GTR + G + I, both DsMV isolates were on a distinct clade, along with an isolate of DsMV from the Cook Islands (Figure 3D).

A contig with 2.3 kb in length (970 reads, 56× coverage) was identified in the HTS dataset that putatively possessed a partial ORF encoding a protein product with moderate homology to the RNA-dependent RNA polymerase (RdRp) of umbra-like viruses (Table 1). Efforts to retrieve additional sequence information on this agent, tentatively designated CULV, both in silico (iterative mapping and extension of the original contig) and in vitro (RT-PCR and RACE using polyadenylated cDNA and dsRNA) were unsuccessful. The CULV sequence was only identified in a taro sample from Kokoda and deposited in GenBank as CULV-PNG-K (ON086742). Phylogenetic analysis using the RdRp-inferred sequence and GTR + G + I placed this contig on a distinct branch within the umbra-like virus clade. This further supported its existence as a novel virus or virus-like agent (Figure 3E).

The full genome of TaRV identified in two samples from Popondetta, has been reported elsewhere [33]. Briefly, the complete genome of TaRV was comprised by ten dsRNA segments that ranged from 1168 to 3878 bp (MW148346-MW148355) and presented between 16.4 to 54.6% protein identities to homologs coded by the oryzavirus rice ragged stunt virus (RRSV). These low protein identities, lack of serological relationship with RRSV, and phylogenies inferred of TaRV with other oryzaviruses and reovirids suggested that TaRV is a new member of the genus *Oryzavirus* (*Reoviridae* family) [33].

## 4. Discussion

Despite being a severe disease of taro, ABVC remains understudied. Fundamentally, critical knowledge gaps on its etiology, host-pathogen interactions, and transmission remain to be studied. Most examinations of the disease were performed by electron microscopy, which has a limited ability to identify the agents present. In the last three decades, several viruses infecting taro have been molecularly characterized and named. But work aimed at identifying the virus(es) associated with ABVC has had limitations, especially if ABVC symptoms are caused by multiple viruses.

### 4.1. CBDaV Is Involved with ABVC

In a survey of taro viruses in Pacific Island countries using PCR-based diagnostic assays, Revill and collaborators [12] found CBDaV (then called CBDV) in 21 plants from PNG and the Solomon Islands affected by ABVC, but not in 72 asymptomatic plants from the same countries. Higgins and collaborators [8] identified and characterized the complete genome of CBDaV from a taro plant with symptoms of bobone. In our study, although a limited number of plant samples were analyzed, CBDaV was the only pathogen found in all symptomatic but no asymptomatic plants. Taken together, PCR-based assays in two separate studies and HTS in our study point to the absolute association between CBDaV infection and ABVC and strongly suggests its involvement in the etiology of this disease. Further biological and transmission assays are required to confirm the involvement of CBDaV in the etiology of ABVC.

### 4.2. Are Other Viruses Involved with ABVC?

The question still remains whether ABVC is indeed a “virus complex” resulting in a synergistic effect of virus co-infection. Viral synergisms in plants cause more severe symptoms than infections caused by either virus alone. These are described in several well-studied diseases including sweet potato virus disease and maize lethal necrosis [34,35]. Early studies using electron microscopy consistently identified multiple virions in ABVC-symptomatic plants [36], leading to a hypothesis of viral synergism. Of the 21 ABVC-affected plants studied by Revill and collaborators [12], 19 were co-infected with other viruses, most notably TaBV, which was present in 15 (71%) of the affected plants. By using virus-specific primers in PCR-based assays, however, the status of viruses undescribed at the time of the study, such as TaBCHV, were unknown. Furthermore, variant genotypes of the targeted viruses might not have been detected in some samples. HTS was used to characterize the genome of CBDaV [8], a technique that can overcome the limitations of virus-specific PCR assays for virus indexing, but it was not reported if viruses other than CBDaV were present. In our study, we identified TaBCHV in all ABVC-affected plants. Taken together, if another virus is involved in the etiology of ABVC, the most likely candidate would be a badnavirus, either TaBV or TaBCHV. In our study, we corroborated episomal badnavirus presence using (DNase)-RT-PCR and RCA assays in all the samples collected from PNG (Appendix A). Bioinformatic analysis (iterative mapping) allowed us to obtain a circular scaffold hallmarking the TaBV episomal genome in one taro plant affected by ABVC in Kokoda-PNG (Appendix A). Although Revill and collaborators [12] identified six ABVC-affected plants that were TaBV negative, all of which were from PNG, it is possible these plants were co-infected by TaBCHV. In our study, all five plants from three separate locations were TaBCHV-positive, suggesting the virus is widespread in the country.

The involvement of badnaviruses in synergistic diseases is well-documented [37], so the theory that ABVC requires a co-infection with CBDaV and a badnavirus is plausible and has been suggested before using electron microscopy [1,2,4,5]. Conversely, it is also possible that CBDaV is the only agent involved in ABVC and any co-infecting viruses contribute to the symptomology, but not in a synergistic manner. The presence of badnaviruses in ABVC-affected plants may just be a result of their relatively high incidence in taro-growing areas of the Pacific. Yang and others [9], and Revill and collaborators [12] respectively found overall incidences of 73 and 60% for TaBV in the region, but a more recent study [38] reported an incidence of 73% for TaBCHV in Hawaii. It is evident that controlled biological assays using virus-free taro plants are required to completely understand the etiology of ABVC. The permit conditions for sample importation and storage in RNAlater solution^®^ prevented our ability to conduct such experiments for our study. Such assays could involve the use of vector-mediated transmission using planthoppers for CBDaV [8] and mealybugs for badnaviruses [39]. Alternatively, badnavirus infection can be established using an infectious clone as has been previously reported for an Australian isolate of TaBV [40]. Following the genetic strategies for development of plant rhabdovirus infectious clones [41] and the agroinoculation strategy detailed for the TaBV infectious clone [40], an infectious clone of CBDaV represents an elegant strategy to evaluate the role of the virus in the etiology of ABVC.

Prior to this study, genomic sequence information was only available for one isolate of CBDaV. By sequencing two additional isolates, we have documented some of the genetic diversity of this virus, which is essential for the development of reliable molecular detection assays. These sequence data also confirmed the presence of a small ORF between those encoding the G and L proteins reported in a recent study on a newly characterized strawberry cytorhabdovirus [42]. In addition to the genetic diversity documented for CBDaV, we also identified genetic variants of other taro-infecting viruses.

Using these data, we were able to develop new molecular detection assays for each virus that may help reduce type II errors (false negatives) during diagnostic testing. Given the genetic variation of viruses infecting taro: CBDaV up to 17%, this study, TaBV up to 14% [43], TaBCHV up to 20% [43], and DsMV up to 23% [44], the presence of taro viruses in asymptomatic material [12], and the regional movement of taro germplasm in the Pacific, makes robust diagnostics critical. Furthermore, our HTS data provided evidence of a novel agent, CULV, in one plant from Kokoda. This indicates there are additional viral or viral-like pathogens of taro yet to be characterized and evaluated for their impact on plant health and productivity.

## Figures and Tables

**Figure 1 viruses-14-01410-f001:**
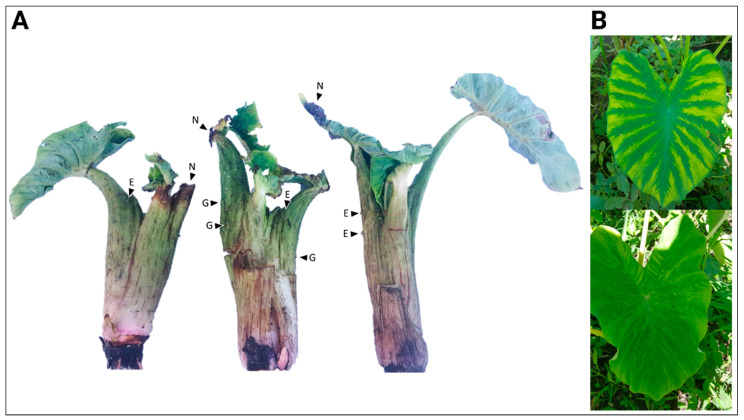
Symptoms of taro plants with and without ABVC symptoms. (**A**) Symptoms of the three taro plants with ABVC symptoms collected from Kokoda and Popondetta in Papua New Guinea (PNG). E = enations; N = necrosis; G = galls. (**B**) Two taro plants sampled from Girua, PNG presenting symptoms of interveinal chlorosis, but not ABVC.

**Figure 2 viruses-14-01410-f002:**
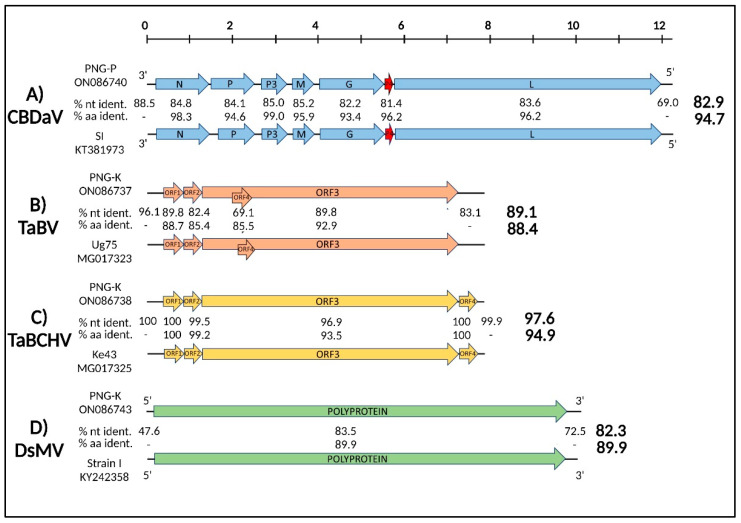
Genomic organization and nucleotide and protein identities of representative virus isolates from Papua New Guinea (PNG) of Colocasia bobone disease-associated virus (CBDaV) (**A**), taro bacilliform virus (TaBV) (**B**), taro bacilliform CH virus (TaBCHV) (**C**), and dasheen mosaic virus (DsMV) (**D**) with their closest homologs. Open reading frames (ORFs) are represented by light-colored arrows. A new proposed ORF (P4) found in CBDaV in two isolates from PNG and one isolate from the Solomon Islands is in red. The overall nucleotide and protein identities are located to the right of each virus genome representation. The virus strain names and their respective GenBank accessions are provided. Genomes and ORFs are to scale.

**Figure 3 viruses-14-01410-f003:**
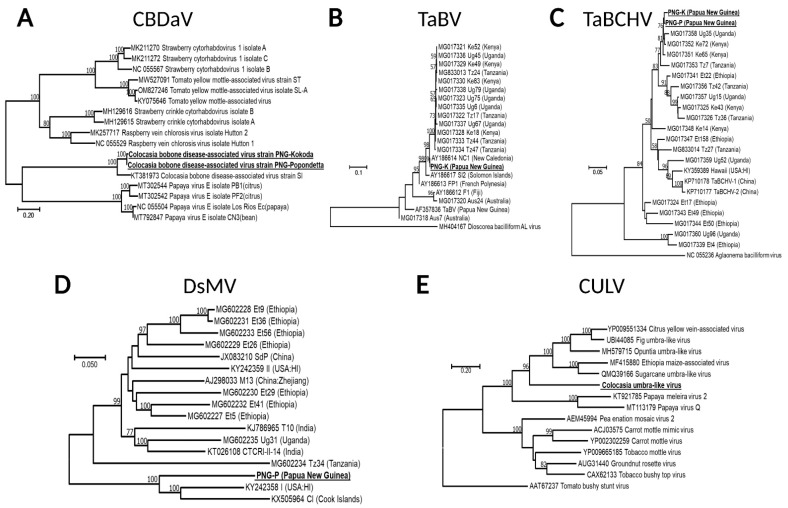
Phylogenetic relationships of CBDaV, TaBV, TaBCHV, DsMV, and CULV with their closest homologs. Maximum likelihood algorithm with 1000 bootstrap repetitions as branch support was used to infer phylogenies of nucleotide or amino acid sequences and their best model of evolution. (**A**) CBDaV (whole genome sequence, LG + G + I), (**B**) TaBV and (**C**) TaBChV (partial nucleotide sequence, GTR + G + I); (**D**) DsMV (whole genome sequence, GTR + G + I), (**E**) CULV (RdRp protein sequence, GTR + G + I).

**Table 1 viruses-14-01410-t001:** Number of high-throughput sequencing (HTS) reads, contigs, and depth of coverage values for each virus found in ABVC-symptomatic taro plants from Papua New Guinea.

No. Raw Reads	Non-Host Reads	Virus ^a^	Contigs Number	Genome Length (nt) ^b^	Mapped Reads ^c^	Coverage (No. Reads) ^c^		
						Min.	Mean	Max
14,533,374	8,389,201	CBDaV	61	12,205	184,663	2	2296	42,272
		TaBV	16	7825	2,542,726	26	137,637	1,287,793
		TaBCHV ^d^	37	7769–7770	2176	2	24	588
		TaRV ^c^	55	1168–3878	4,780,464	1–4	10,177–59,941	25,379–287,699
		DsMV	18	10,066	3902	2	46	859
		CULV	1	2360	970	1	56	344

^a^ RNA viruses: CBDaV, Colocasia bobone disease-associated virus; TaRV, taro reovirus; DsMV, dasheen mosaic virus; CULV, Colocasia umbra-like virus. DNA viruses: TaBV, taro bacilliform virus; TaBCHV, taro bacilliform CH virus. ^b^ Genome length corresponds to the final complete genome sequence of all taro viruses, except for CULV that a partial sequence is reported. The complete genomes were obtained using HTS, RT-PCR assays, RACE and Sanger sequencing. ^c^ Mapped reads and coverage correspond to the number of non-host reads that mapped to the complete and partial genome sequences. TaRV has 10 genomic segments and therefore a variable number of reads mapped to each genomic sequence of the virus. ^d^ Two complete genomes of TaBCHV were obtained in this study from two ABVC symptomatic plants and presented ~97% nucleotide identity between each other (7769 and 7770 nts).

**Table 2 viruses-14-01410-t002:** Virus indexing in taro plants with and without symptoms of ABVC collected from three locations, Popondetta, Kokoda, and Girua, in Papua New Guinea using RT-PCR assays and primer sets detailed in Appendix A. Full virus names are found in the main text.

Sample	Location	ABVC Symptoms	CBDaV	TaBCHV	TaBV	TaRV	DsMV	CULV	TaVCV
1	Kokoda	Yes	+	+	+	−	+	+	−
2	Popondetta	Yes	+	+	−	+	−	−	−
3	Popondetta	Yes	+	+	−	+	−	−	−
4	Girua	No	−	+	+	−	−	−	−
5	Girua	No	−	+	−	−	−	−	−

## Data Availability

All the sequence data that was generated in this study has been uploaded to the GenBank database.

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
