# Peer review of "Examination of the Virome of Taro Plants Affected by a Lethal Disease, the Alomae-Bobone Virus Complex, in Papua New Guinea"

_viruses, 2022, doi:10.3390/v14071410_

Round 1

Reviewer 1 Report

The paper presents interesting results expanding the knowledge on understudied taro viruses. The advantage of the paper is that it combines the results of different detection methods that together support the conclusions. Because of the different methods used, it is important to clearly present the results (from which sample they originate from and what method was used) and I believe this can be improved. I have some concerns about the low number of plant samples included in the study, therefore the involvement of detected viruses in the etiology of the disease should be carefully discussed. I do not believe that any additional experiments or analysis need to be performed, however the selection of certain methods and analysis steps should be explained and supported by relevant references.

Specific comments to improve the manuscript:

In the introduction part:

-references regarding disease symptoms and geographic distribution of this disease are missing

-explain that BBDaV was previously called CBDV in the introduction part

-flexuous rods (line45) – change to flexuous rod-shaped viral particles for better understanding

-binominal nomenclature is used for virus species in some cases, but not for all of the presented viruses – please standardize the virus species names (in introduction and the results section ) if possible

- type II error should be shortly explained as false negative (also in line 321)

-results of Revill and collaborators (line 54) are somehow missing – describe the main conclusions of this study or do not mention this reference at all in the introduction

M&M part:

-Figure 1 and the 2.1 paragraph: make a clear connection between the Figure 1 and 2.1 paragraph

-Figure 1: A) 2xsymptoms, B) there are two pictures – what is the difference between them?

-2.1 paragraph: Symptoms of interveinal chlorosis are only mentioned below the figure – explain why you decided to samples this two plants. Cite Figure 1 also for plants with no ABVC-related symptoms.

-dsRNA was used for sequencing – Please explain why did you decide to sequence dsRNA and not for example CTAB isolated TNAs.

-episomal and integrated form of badnaviruses were previously reported. You decided to study the episomal forms, however in your bioinformatics analysis you mapped the sequencing reads to the reference genome and in this way forgetting about the potential integrated forms of viruses. I do not know the completeness of the host reference genome used, however I believe the discussion about episomal and integrated forms of badnaviruses together with the effect/bias of your methodological steps is missing.

-the description of the sequencing of amplicons and amplicons cloned into pGEM-T Easy is missing in the M&M section

-From M&M is looks like RT-PCR was done on dsRNA and PCR, RT-PCR and DNAse-RT-PCR were done on TNAs for all the detected viruses… Some results are presented in Figure S1 and it is not clear how the Table 2 was prepared. In the cases where PCR was used to validate the HTS output it is necessary to show all the results in the Supplementary materials.

-information on how did you determine the best model of evolution is missing in the M&M section

Results part:

-In the 3.2 paragraph clearly state which samples were sequenced (just like you do for PCR-based assays in line 165). It is clear from the M&M that the dsRNA were extracted from pooled ABVC-symptomatic leaf tissues, however the title of Table 1 is introducing some confusion with linking the HTS and non-symptomatic plants.

-Table 1: For every column of the table define if the results are based on HTS (reads, contigs number) or some other methods used. For “Genome length” it is not clear if the number represents the longest contig sequence or it is based on mapping to the reference sequence. Explain why you report a range of numbers for TaBCHV and TaRV genome length and for TaRV also a range for coverage values. In TaBCHV genome length – correct the number 77770. Clarify the mentioning of non-symptomatic plants (that were not sequenced) in the title of Table 1.

-genome and proteome sequence (line 192 and others) – correct to nucleotide and amino acid/protein sequences.

-Line 193 – 82.9 and 94.7% identity with the PNG-isolates of the virus – correct to PNG-P isolate

-Table 2: Define which methods you used for virus indexing presented in this table. Can this results be supported by gel electrophoresis of PCR amplicons? Why is there a difference in Figure S1 for TaBCHV detection (PCR vs. RT-PCR for sample P2)?

-Figure 2: Correct the use of commas in the title. Clearly and in the same way for all viruses present (maybe without abbreviations) the strain names, country of origin and GenBank accessions. Please include the scale for the presented genome lengths.

-describe the methods (e.g. RACE) used for additional characterization of umbra-like sequence in the M&M sections prior to mentioning them in the results section.

-TaRV – full genome has been reported elsewhere, however you could briefly report on the detected sequences and their GenBank accessions in line with the results described for other detected viruses.

Discussion part:

-in the part where you discuss the association between BCDaV and ABVC the low number of analyzed plant samples should be emphasized.

-I believe your discussion about the need for controlled biological assays using virus-free taro plants is very relevant. You could even strengthen this part and briefly describe what would such experiments (prevented by the permit conditions) look like.

-how did you determine the genetic variation of CBDav (18%) and is this number comparable to numbers reported for other taro viruses (TaBV, DsMV…) from different studies?

Author Response

The paper presents interesting results expanding the knowledge on understudied taro viruses. The advantage of the paper is that it combines the results of different detection methods that together support the conclusions. Because of the different methods used, it is important to clearly present the results (from which sample they originate from and what method was used) and I believe this can be improved. I have some concerns about the low number of plant samples included in the study, therefore the involvement of detected viruses in the etiology of the disease should be carefully discussed. I do not believe that any additional experiments or analysis need to be performed, however the selection of certain methods and analysis steps should be explained and supported by relevant references.

- Dear reviewer #1, we appreciate the time, effort and feedback you have provided to improve the clarity of our manuscript. We have provided point-by-point responses to your concerns and have highlighted where corrections were implemented when needed.

Specific comments to improve the manuscript:

In the introduction part:

-references regarding disease symptoms and geographic distribution of this disease are missing

- The references 1 through 4 in our manuscript provide enough background on the disease, ABVC and the few studies of viral agents associated with it. Based on these references cited in our manuscript, we have added a few edits so that it is clearly stated the disease is limited to the Solomon Islands and Papua New Guinea. All this information and disease symptoms can be found in L36-43.

-explain that CBDaV was previously called CBDV in the introduction part

We have mentioned that CBDaV was previously called CBDV, please check L46-47 and L59-61.

-flexuous rods (line45) – change to flexuous rod-shaped viral particles for better understanding

change done, please check L49

-binominal nomenclature is used for virus species in some cases, but not for all of the presented viruses – please standardize the virus species names (in introduction and the results section ) if possible

We have used binomial nomenclature on the virus species (Cytorhabdovirus colocasiae for CBDaV and Alphanucleorhabdovirus colocasiae for TaVCV) for which it has been officialized as available in the ICTV website. No official binomial nomenclature has been implemented for potyviruses (DsMV) or badnaviruses (TaBV and TaBCHV). Since umbra-like viruses are not yet officially recognized as a virus taxon (therefore called umbra-like viruses) and no official binomial nomenclature has been implemented for umbraviruses, we designated the new umbra-like virus in taro as Colocasia umbra-like virus (CULV).

- type II error should be shortly explained as false negative (also in line 321)

Thanks for the suggestion. This has been implemented in L62 and L354.

-results of Revill and collaborators (line 54) are somehow missing – describe the main conclusions of this study or do not mention this reference at all in the introduction

Additional information on the main conclusions of Revill & collaborators (2005) has been added to our manuscript, please check L59-61.

M&M part:

-Figure 1 and the 2.1 paragraph: make a clear connection between the Figure 1 and 2.1 paragraph

Changes have been implemented in 2.1 paragraph, please check L79 and L87.

-Figure 1: A) 2xsymptoms, B) there are two pictures – what is the difference between them?

Figure 1A represents the three plants presenting ABVC symptoms in PNG while Figure 1B depicts the two plants presenting interveinal chlorosis, but not ABVC symptoms. Edits were added to the title of Figure 1 to clearly explain Figures 1A and 1B, please check L115-118.

-2.1 paragraph: Symptoms of interveinal chlorosis are only mentioned below the figure – explain why you decided to sample these two plants. Cite Figure 1 also for plants with no ABVC-related symptoms.

Changes have been implemented in 2.1 paragraph, please check L79 and L87. Also, these plants presenting interveinal chlorosis were sampled as part of another unrelated taro project on TaVCV (although no TaVCV was found on these plants). The description of the no-ABVC related symptoms (interveinal chlorosis) is detailed in Paragraph 3.2 (L163-164).

 -dsRNA was used for sequencing – Please explain why did you decide to sequence dsRNA and not for example CTAB isolated TNAs.

Some edits were added to explain the rationale on the use of dsRNAs as starting material for HTS to identify, diagnose and characterize plant viruses, please check L89-91.

-episomal and integrated form of badnaviruses were previously reported. You decided to study the episomal forms, however in your bioinformatics analysis you mapped the sequencing reads to the reference genome and in this way forgetting about the potential integrated forms of viruses. I do not know the completeness of the host reference genome used, however I believe the discussion about episomal and integrated forms of badnaviruses together with the effect/bias of your methodological steps is missing.

We have added some edits to emphasize that our in silico results point to the presence of an episomal TaBV form in the taro plant affected by ABVC in Kokoda-PNG, please check L189-191 and 317-321.

Integrated forms of badnaviruses have been demonstrated for some virus members. In the case of TaBV and TaBCHV it has been speculated but not demonstrated that these viruses may be integrated into the taro genome. Still, if taro badnavirus(es) is(are) integrated into the taro genome, the virus sequences would be integrated as randomly disordered arrays of sequences (such as those reported for BSV and other badnaviruses). We are confident that our bioinformatics analysis (Figure S1C) shows that a circular scaffold was obtained after the iterative mapping step (highlighted by the white arrows). Furthermore, HTS reads mapped to the linearized TaBV genome are in an orderly fashion with no presence of ‘chimeric’ HTS sequences that would originate from virus integrated forms into the taro genome. These latter results were not included in the manuscript/figures since they are redundant with Figure S1C.

-the description of the sequencing of amplicons and amplicons cloned into pGEM-T Easy is missing in the M&M section

We included the descriptions of the direct sequencing or cloning into pGEM-T-Easy and sequencing of at least 3 clones in materials and methods in sections 2.2 and 2.3 of our original manuscript.

-From M&M is looks like RT-PCR was done on dsRNA and PCR, RT-PCR and DNAse-RT-PCR were done on TNAs for all the detected viruses… Some results are presented in Figure S1 and it is not clear how the Table 2 was prepared. In the cases where PCR was used to validate the HTS output it is necessary to show all the results in the Supplementary materials.

Some edits were added to clarify this concern, please check L110-110. RT-PCR assays to validate HTS output and bridge sequence gaps were performed using either dsRNAs (extracted from pool of tissue of the three ABVC samples) or TNAs extracted from the five individual samples used in our study (3 presenting ABVC symptoms and 2 presenting other viral-like symptoms not related to ABVC). Also, a supplementary Figure 2 has been uploaded with the revised version of this manuscript which includes agarose gel electrophoresis that shows the amplicons of the RT-PCR assays for detection of the taro viruses detailed in Table 2.

-information on how did you determine the best model of evolution is missing in the M&M section

Information on how the best model of DNA/protein evolution was determined has been added, please check L154.

 Results part:

-In the 3.2 paragraph clearly state which samples were sequenced (just like you do for PCR-based assays in line 165). It is clear from the M&M that the dsRNA were extracted from pooled ABVC-symptomatic leaf tissues, however the title of Table 1 is introducing some confusion with linking the HTS and non-symptomatic plants.

Thank you for pointing this out. We have added some edits to paragraph 3.2 and the title of table 1 to clarify this, please check L167-8 and L194-6.

-Table 1: For every column of the table define if the results are based on HTS (reads, contigs number) or some other methods used. For “Genome length” it is not clear if the number represents the longest contig sequence or it is based on mapping to the reference sequence. Explain why you report a range of numbers for TaBCHV and TaRV genome length and for TaRV also a range for coverage values. In TaBCHV genome length – correct the number 77770. Clarify the mentioning of non-symptomatic plants (that were not sequenced) in the title of Table 1.

Some edits have been implemented as per your suggestion in Table 1, please check.

-genome and proteome sequence (line 192 and others) – correct to nucleotide and amino acid/protein sequences.

Change made, please check L205, 225, 236, 246, and 255.

-Line 193 – 82.9 and 94.7% identity with the PNG-isolates of the virus – correct to PNG-P isolate

Both PNG isolates of CBDaV (PNG-K and PNG-P) and their protein-inferred sequences display 82.9 and 94.7% nucleotide and protein identities, respectively, to CBDaV from Solomon Islands. Therefore, the statement in L205 and information in Figure 2A are both accurate.

-Table 2: Define which methods you used for virus indexing presented in this table. Can this results be supported by gel electrophoresis of PCR amplicons? Why is there a difference in Figure S1 for TaBCHV detection (PCR vs. RT-PCR for sample P2)?

Edits were added to the title of table 2 to clarify virus indexing was performed using RT-PCR assays and virus-specific primer sets detailed in table S1. We are uncertain onto why DNA template (as part of the total nucleic acids) from sample P2 tested negative for TaBCHV using PCR assays but tested positive using RT-PCR assays (with or without DNase step). Importantly, the same PCR result was obtained for this sample when the PCR assay was repeated twice. A congruent result on this line was obtained in the RCA assays since faint amplicons were observed in an agarose gel electrophoresis for this sample only (Figure S1B). It is possible that TaBCHV DNA genome titration in sample P5 is very low and undetectable using PCR. However, TaBCHV RNA transcripts in the same sample may be more abundant and enriched in the cDNA synthesis step so that they can be detected by RT-PCR assays. Regardless, in our manuscript, we emphasize the use of RT-PCR assays and (DNase)-RT-PCR assays for confirmation of episomal badnavirus infection. Additionally, a supplementary Figure 2 has been uploaded with the revised version of this manuscript which includes agarose gel electrophoresis that show the amplicons of the RT-PCR assays for detection of the taro viruses detailed in Table 2, except for TaBV and TaBCHV whose agarose gel electrophoresis pictures are in supplementary Figure 1.

-Figure 2: Correct the use of commas in the title. Clearly and in the same way for all viruses present (maybe without abbreviations) the strain names, country of origin and GenBank accessions. Please include the scale for the presented genome lengths.

Thanks for pointing this out. Some edits were added to Figure 2 and the title of Figure 2, please check

-describe the methods (e.g. RACE) used for additional characterization of umbra-like sequence in the M&M sections prior to mentioning them in the results section.

The suggestion is appreciated. However, we consider that the methods used to further characterize CULV are clearly described in M&M section, i.e. iterative mapping (in silico) and RT-PCR and RACE (in vitro). We consider that adding the extra efforts that we adopted to extend the genome sequence of CULV to M&M would provide in advance some results of our study. Therefore, this information was located in the results section 3.3.

-TaRV – full genome has been reported elsewhere, however you could briefly report on the detected sequences and their GenBank accessions in line with the results described for other detected viruses.

Some edits were added that briefly detail TaRV genome, protein identities to its closest homolog as well as extra information that shows TaRV is a new member classified within the Oryzavirus genus (Reoviridae family). Please, check L271-6

Discussion part:

-in the part where you discuss the association between CBDaV and ABVC the low number of analyzed plant samples should be emphasized.

Paragraph 4.1 has been modified, please check L295-300.

-I believe your discussion about the need for controlled biological assays using virus-free taro plants is very relevant. You could even strengthen this part and briefly describe what would such experiments (prevented by the permit conditions) look like.

Some edits were added as per your suggestion, please check L339-45.

-how did you determine the genetic variation of CBDav (18%) and is this number comparable to numbers reported for other taro viruses (TaBV, DsMV…) from different studies?

Apologies, this number should be 17%, since the lowest nucleotide identity value between CBDaV isolates is 82.9% (Figure 2). Perhaps we misunderstand the question, but we think comparing the nucleotide diversity values for very different virus taxa is not informative; some virus groups exhibit much higher diversity than others. For example, closteroviruses will have much greater sequence variability within isolates of a species than tobamoviruses. Our purpose for reporting these rather high diversity values is to illustrate both the difficulty and importance of developing of robust PCR-based detection assays.

Reviewer 2 Report

In this study, Olmedo-Velarde et al. investigated the virome of taro plants from to location in Africa. The overall approach of this study was to conduct high throughput sequencing using double-stranded RNAs from the collected samples. The authors report a mixed infection of RNA and DNA viruses. Also, the authors report the presence of a putative novel virus in taro plants. Please see below my comments and suggestions:

Abstract: In general, I would recommend providing a little background about this study. Abstract needs to be restructured giving a more easy-to-follow flow including brief background, brief materials and methods, and meaningful results.

Page 1, line 38: Use "Figure 1A" because galls are shown in panel 1A.

Page 1, line 41: To make this statement clearer, I recommend adding "After electron microscopy analyses, ABVC was initially associated with..."

Page 2, line 56: Please be more clear/specific with the statement "type II errors."

Page 2, line 75: The exact total of samples is vague. Please mention how many samples were used for HTS including the additional leaf samples that were not related with ABVC symptoms.

Page 2, line 82: Please provide a reasoning of why dsRNAs were used instead of other type of nucleic acid (i.e., total RNA).

Page 3, line 105: Figure 1 has three panels; hence, I suggest using "C" on the lower right photo of the figure.

Page 4, line 126: It is assumed that these TNAs were not treated with DNase. If that is the case, please state that in the description to secure clarification.

Page 4, line 133: I suggest using "In Silico Genomic and Proteomics Analyses" because based on your description, you only used the available platforms to analyze the sequences.

Page 5, line 181: Indicate in table 1 the type of virus based on nucleic acid (DNA or RNA virus).

Page 6, line 224: This sentence fits better in Materials and Methods.

Page 7, line 242: Accessions numbers are not provided in figure 2 as indicated as indicated.

Page 8, line 256: In figure 3, indicate which trees were analyzed with either nucleotide or amino acid sequences.

Page 9, line 281: Please use "in" instead of "ins."

Page 9, line 302: This statement is not clear. Earlier you mention that synergistic interactions greatly contribute to symptomatology, here, it sounds confusing and contradictory at the same time. 

Page 9, line 321: Please be more clear/specific with the statement "reduce type II errors."

Author Response

In this study, Olmedo-Velarde et al. investigated the virome of taro plants from to location in Africa. The overall approach of this study was to conduct high throughput sequencing using double-stranded RNAs from the collected samples. The authors report a mixed infection of RNA and DNA viruses. Also, the authors report the presence of a putative novel virus in taro plants. Please see below my comments and suggestions:

Dear reviewer #2, we appreciate the time, effort and feedback you have provided to improve the clarity of our manuscript. We have provided point-by-point responses to your concerns and have highlighted where corrections were implemented when needed.

Abstract: In general, I would recommend providing a little background about this study. Abstract needs to be restructured giving a more easy-to-follow flow including brief background, brief materials and methods, and meaningful results.

Some edits were implemented in the abstract, please check.

Page 1, line 38: Use "Figure 1A" because galls are shown in panel 1A.

Change done, thanks for the suggestion.

Page 1, line 41: To make this statement clearer, I recommend adding "After electron microscopy analyses, ABVC was initially associated with..."

Change done, thanks for the suggestion.

Page 2, line 56: Please be more clear/specific with the statement "type II errors."

Thanks for the suggestion. This has been implemented in L62 and L354 of the revised manuscript.

Page 2, line 75: The exact total of samples is vague. Please mention how many samples were used for HTS including the additional leaf samples that were not related with ABVC symptoms.

Some edits were added to clarify the number of leaf samples collected from ABVC-symptomatic plants as well as leaf samples displaying viral-like symptoms not related to ABVC (L79-87). HTS was performed from a cDNA library prepared from dsRNA extracted exclusively from ABVC-symptomatic leaf tissue as detailed in lines 89 – 96.

Page 2, line 82: Please provide a reasoning of why dsRNAs were used instead of other type of nucleic acid (i.e., total RNA).

Some edits were added to explain the rationale on the use of dsRNAs as starting material for HTS to identify, diagnose and characterize plant viruses, please check L89-91.

Page 3, line 105: Figure 1 has three panels; hence, I suggest using "C" on the lower right photo of the figure.

The suggestion is appreciated; however, we consider that “Figure 1A” and “Figure 1B” depicts the two types of leaf samples coming from plants presenting or not ABVC symptoms, respectively. Figure 1A displays the three plants presenting ABVC symptoms while Figure 1B (both pictures/panels) display the two plants presenting interveinal chlorosis that is not related to ABVC.

Page 4, line 126: It is assumed that these TNAs were not treated with DNase. If that is the case, please state that in the description to secure clarification.

Thanks for pointing this out. Some edits were added to clarify this, please check L136.

Page 4, line 133: I suggest using "In Silico Genomic and Proteomics Analyses" because based on your description, you only used the available platforms to analyze the sequences.

Thanks for the suggestion which was implemented. Please, check L144-5.

Page 5, line 181: Indicate in table 1 the type of virus based on nucleic acid (DNA or RNA virus).

Some edits have been made to table 1, please check.

Page 6, line 224: This sentence fits better in Materials and Methods.

We appreciate the suggestion. However, we consider this is also appropriate for the results section considering different models of DNA/protein evolution were used for inferring the phylogenies of the taro viruses in our study. The addition of all the different models of DNA/protein evolution that were used to the Materials and Methods section may make that section too wordy.

Page 7, line 242: Accession numbers are not provided in figure 2 as indicated.

Thanks for pointing this out. This mistake has been corrected in the new figure 2, please check.

Page 8, line 256: In figure 3, indicate which trees were analyzed with either nucleotide or amino acid sequences.

Thanks for the suggestion, some edits were added to the title of figure 3 to clarify which trees were analyzed with either nucleotide or amino acid sequences.

Page 9, line 281: Please use "in" instead of "ins."

Change done

Page 9, line 302: This statement is not clear. Earlier you mention that synergistic interactions greatly contribute to symptomatology, here, it sounds confusing and contradictory at the same time.

We have deleted the sentence “Both CBDaV and TaRV are likely vectored by planthoppers [8,33]” to keep the line in the narrative of that paragraph. Since multiple viruses were found in the ABVC-affected plants, we are hypothesizing that perhaps viral synergism may be causing the detrimental and lethal symptoms of the disease. However, proper assays must be performed to corroborate the synergism: increase of titration of either CBDaV or badnavirus. Therefore, in L326-345 of the revised version of the manuscript we are just exploring all the possible scenarios that may be involved in the etiology of ABVC. Since we were limited to perform molecular assays only, we do not have the permits for performing biological assays, we, therefore, propose the possible scenarios. Also, we have added more info about such biological assays we were not able to implement, please check L339-345.

Page 9, line 321: Please be more clear/specific with the statement "reduce type II errors."

Thanks for the suggestion. This has been implemented in L62 and L354 of the revised manuscript.